# Effects of peer tutoring on middle school students' mathematics self-concepts

Lidon Moliner[1]*, Francisco Alegre[2]

**1** Department of Education, Jaume I University, Castellon, Spain, **2** Department of Education, Valladolid University, Valladolid, Spain

☯ These authors contributed equally to this work.

* mmoliner@uji.es

**Data Availability Statement:** Data are held in a public repository (OSF Storage). doi: 10.17605/OSF.IO/RK43G.

**Funding:** The authors received no specific funding for this work.

## Abstract

The effects of peer tutoring on students' mathematics self-concepts were examined. The Marsh questionnaire was used to measure students' mathematics self-concepts before and after implementation of a peer tutoring program. A pretest posttest control group design was employed. Study participants included 376 students from grades 7 to 9 (12 to 15 years old). No statistically significant differences were reported between the pretest and the posttest for any of the control groups. Statistically significant improvements were reported for all grades for the experimental groups. An average increment of 13.4% was reported for students in the experimental group, and the overall effect size was reported to be medium (Hedges' $g$ = 0.48). No statistically significant differences were reported across grades for the experimental group. The main conclusion of this study is that same-age and reciprocal peer tutoring may be very beneficial for middle school students' mathematics self-concepts. Several recommendations for field practitioners emanated from the study: use same-age and reciprocal tutoring over cross-age and fixed peer tutoring; schedule tutoring programs for four weeks or less with two to four sessions of 25 minutes or less per week for each tutoring session; and, include a control group in research studies.

## Introduction

Peer tutoring may be defined as a flexible teaching strategy in which half of the students serve as academic tutors and the other half serve as academic tutees [1]. In this methodology a higher achieving student (tutor) provides assistance with academic content to a lower achieving student (tutee). Several benefits have been documented across the literature for both tutors and tutees during peer tutoring experiences. From an academic perspective, the majority of studies report significant improvements in the students' mathematics scores [2, 3]. The social implications of peer tutoring are also valuable, as it fosters student inclusion [4] and improves the class climate [5]. Different psychological variables have been widely addressed in the field, such as anxiety or attitude towards mathematics [6, 7, 8, 9]. Most studies reporting promising results for academic, social, and psychological variables showed close to medium effect sizes [10]. One of the variables that has yet to be thoroughly addressed is self-concept. Results from

**Competing interests:** The authors have declared that no competing interests exist.

previous studies in the field are inconclusive. While some authors indicate that peer tutoring may be beneficial and that their students showed improvements [11, 12], others state that no significant improvements or benefits were documented [13]. Hence, given the proven potentiality of these methodologies with academic and psychological variables, it is of interest to test the effects of peer tutoring on students' self-concept.

## Theoretical framework

### Self-concept

Several authors have defined self-concept through the years. Marsh et al. (2019) [14] defined self-concept as individuals' self-perceptions formed through their own experiences and interpretations of their environment. According to Parker, Marsh, Guo, Anders, Shure, and Dicke (2018) [15], an alternative definition is the sum total of individuals' mental and physical characteristics and their own evaluation of them. Hence, self-concept is believed to have a complex structure with different factors comprising it. In this sense, Marsh et al. (2018) [16] identified three main aspects of self-concept: the behavioural (action), the affective (feeling), and the cognitive (thinking). The importance of this psychological variable has been broadly documented. Susperreguy, Davis-Kean, Duckworth, and Chen (2018) [17] Walgermo, Frijters, and Solheim (2018) [18] and Wolff, Nagy, Helm, and Möller (2018) [19] authored papers in which they described how self-concept can both predict and influence academic achievement in different subjects, such as mathematics, reading, and literature, across different learning levels, including primary and secondary education. Hence, considering that an increase in students' self-concepts usually results in an increase in their academic achievement [20, 21], it is of interest, from an academic perspective, to examine teaching methodologies that may positively influence students' self-concepts.

### Mathematics self-concept

This research aims directly at students' self-concepts regarding mathematics, which is the academic subject students spent the most time studying [22]. Research by Pajares and Miller (1994) [23] and Pietsch, Walker, and Chapman (2003) [24] documented the high correlation between students' mathematics self-concepts and their mathematics academic achievement from primary education to college. Sax (1994) [25] stated that the direct relationship of this psychological variable with students' academic performance increases as they get older. In this sense, according to Marsh and O'Neill (1984) [26], the structure of the mathematics self-concept is multifaceted and hierarchical with facets becoming more distinct with age. Marsh and Shavelson (1985) [27] and Lee (2009) [28] concluded that mathematics was the subject in which students' academic performance was influenced most by their subject-related self-concept. According to these authors a positive self-concept may help with mathematics performance given the effects that produces in variables such as motivation or on task behavior. If a student truly beliefs he/she can solve a mathematics problem, he/she will have the necessary resilience to persist until he/she can solve it. Students with a high self-concept may see failed attempts as exciting challenges and new opportunities, while students with low self-concept will doubt their own abilities and give up early after few attempts [29, 30]. Besides, authors such as Sticca, Goetz, Bieg, Hall, Eberle, and Haag (2017) [31] or Onetti, Fernández-García, and Castillo-Rodríguez (2019) [32] state that the transition from primary school to middle school usually results in a decrease in students' mathematics self-concept during the first year, that is, 7th grade. Factors such as a higher difficulty in the mathematics contents, substantially more difficult exams, the change of learning environments and methodologies are associated

with this decrease. Given the importance of this variable, several validated instruments have been developed to measure it at different educational levels [33].

## Peer tutoring

Peer tutoring is an active teaching methodology that fosters student inclusion while enabling students to learn from each other [34]. Topping (2018) [35] defined it as a cooperative learning method based on a pairing of students who share learning objectives. These objectives are achieved through a framework in which students have an asymmetric relationship derived from their respective academic competences. In this sense, in each pair one of the students plays the role of tutor and the other plays tutee [36]. Tutees must ask tutors academic questions in order to acquire curriculum content. The main role of the tutors is to provide feedback and help their tutees during the learning process, as tutors, by design, have higher academic competency than their partners [37]. Both tutors and tutees benefit from this methodology. Tutees benefit from receiving direct instruction from a peer. As students share a similar discourse, tutees usually feel more comfortable, ask more questions, and better understand the content [38]. Tutors benefit as they reinforce their knowledge by answering the tutees' questions. These interactions between tutors and tutees promote active learning and foster student inclusion, as all students participate in the process [39, 40].

The different forms that peer tutoring takes depend on a series of factors, with participants' ages and roles being the most important [41]. Experts in the field refer to pairing students of different ages as cross-age tutoring [42]. In this type of tutoring, most often the older student plays the role of tutor. Same-age tutoring have been defined as the pairing of a tutor and tutee of the same age [43]. Same-age tutoring is usually easier to arrange and carry out from an organizational point of view [44]. Depending on the students' roles, fixed or reciprocal peer tutoring may be defined. During reciprocal peer tutoring, students exchange roles, going from tutor to tutee and vice versa. Conversely, in fixed peer tutoring, students do not exchange roles [45]. The benefits of peer tutoring in mathematics have been widely largely documented during the last years [3, 46, 47, 48]. Although psychological variables have not been studied as in depth as academic achievement, several meta-analysis in the field state that the expect effect sizes in a peer tutoring intervention may be considered as small to moderate [49, 50]. Academic and psychological effects of peer tutoring may differ significantly across educational levels. For example, academic effects are usually greater in primary education than in secondary education [47, 48]. Nevertheless, effects within the same educational level are expected to be similar and, when analyzing differences across grades, significant differences are rarely reported [49, 50].

## Peer tutoring and self-concept

Previous meta-analyses in the field by Ginsburg-Block, Rohrbeck, and Fantuzzo (2006) [51] and Ullah, Tabassum, and Kaleem (2018) [52] noted that peer tutoring usually has a positive impact on students' self-concepts, but the significance of the effect has yet to be proven. During the last several years, the latest studies of peer tutoring and students' self-concepts in reading [53], writing [54], English as a foreign language [55], physical education [56], physics [57], and chemistry [58] are promising but far for from being conclusive.

## Peer tutoring and mathematics self-concept

The influence of peer tutoring on students' mathematics self-concepts has been addressed over the last three decades. The pioneer studies by Fantuzzo, King, and Heller (1992) [59], Fantuzzo, Davis, and Ginsburg (1995) [60], Ginsburg-Block and Fantuzzo (1997) [61], and

Topping, Campbell, Douglas, and Smith (2003) [62] focused on the effect of peer tutoring on students' mathematics self-concepts. Half of the studies showed significant improvements regarding self-concepts, while results for the other half were not significant or were inconclusive. Most of them indicated that tutors' self-concept increased significantly after the peer tutoring experience. According to these authors, when a student realizes he/she is able to explain mathematics contents to a peer her/his confidence in his/her own abilities in mathematics increases. In fact, recently, while Zeneli, Tymms, and Bolden (2016) [63] did not find any significant results, Alegre Ansuategui and Moliner Miravet (2017) [2] did find significant improvements regarding mathematics self-concept in this context. Although many articles the research in the field show positive outcomes and improvements in students' mathematics self-concept, only some of them report statistically significant improvements or significant effect sizes. Authors such as Froiland and Davison (2016) [64], Sáinz and Upadyaya, K. (2016) [65], Westphal, Kretschmann, Gronostai and Vock (2018) [66] indicate that peer helping could be beneficial for students mathematics self-concepts and state that more research is needed to address the benefits of peer support in students' emotions. In this sense, the potentiality of peer tutoring with other variables such as mathematics achievement, mathematics anxiety or attitude towards mathematics has been proved. Besides, results found for some studies regarding the mathematics self-concept are inconclusive and there is a need for more literature in the field [67]. Hence, a peer tutoring study that addresses students' mathematics self-concept is performed in this research.

## Methodology

The institutional review board that authorized this research was the Valencian Ministry of Education. They approved the research but the obtained consent specified that data had to be analyzed anonymously.

### Aim of the study and hypotheses

The main aim of this research was to determine the effect of peer tutoring on middle school students' mathematics self-concepts. To this purpose, two hypotheses were defined:

- Hypothesis 1: Students' mathematics self-concepts will improve significantly as a result of peer tutoring.

- Hypothesis 2: No statistically significant differences will be found between $7^{th}$, $8^{th}$ and $9^{th}$ grade students' scores before and after the implementation of the peer tutoring program.

### Research design

Stigmar (2016) [68] stated that the research design employed in a peer tutoring experience may significantly affect the results. According to this author, the absence of a control group or just performing a posttest (pretest posttest without control group and posttest only with control designs) may overestimate the effect of the experience. Hence, following the guidelines given by this author, an experimental pretest posttest with control group design [69] was used in this research so that results were not critically affected by the experimental design.

### Sample access

Difficulty of getting a proper sample in educational research has been discussed by authors such as Kane (2006) [70] or Micklewright, Schnepf and Silva (2012) [71]. In this sense,

students in the study were selected through clustered sampling, which is a sampling technique that divides the population into groups (middle schools, in this case) so that they all share similar characteristics [72]. One public middle school was randomly selected, and students participating in this research were accessed after written informed consent was obtained from their families (parents or guardians of the minors), the School Council, and the Educational Government. Ethical requirements provided by the Ethics Committee of the Spanish National Research Council (CSIC) were met during this research.

### Representativity of the sample

According to the Spanish Educational Government, about 1.5 million students were enrolled in grades 7 to 9 in Spain. The authors of this manuscript sought a study sample representative of the population of middle school students in Spain. According to Krejcie and Morgan (1970) [73] and Johanson and Brooks (2009) [74], at least 368 students were needed to achieve this representation.

### Participants

A total of 376 students, ages 12 to 15 years old, enrolled in grades 7 to 9 participated in this research: 124 were enrolled in 7th grade, 124 were enrolled in 8th grade, and 128 were enrolled in 9th grade. 50.53% were female and 49.47% were male. The average age was 14.21 years old with a standard deviation of 1.37 years. Of the total, 210 (56%) were Hispanic, 94 (25%) were Rumanian, 64 (17%) were African, 4 (1%) were Asian and the other 1% were from other ethnic groups. The socio-economic status of the students' families was average. Students were assigned to the experimental or the control group on a probabilistic basis [75]. Hence, half the students in each course were randomly assigned to the experimental group and the other half to the control group. There were six subgroups in each grade. A draw was performed for each grade so that three subgroups were assigned to experimental conditions and the other three to control conditions. An additional final draw was performed in some subgroups to exclude some students so that the number of students in the experimental group matched the number of students in the control group in each grade [76]. 9 students were randomly excluded due to this procedure.

### Academic content

In the study, the content worked on by students, including algebra, geometry, statistics, and probability, corresponded to the third term of the mathematics courses for each grade. Seventh graders worked with basic first-degree equations, calculated surfaces and volumes of regular prisms, used the Pythagoras' theorem, calculated basic statistical centralization parameters for both qualitative and quantitative variables, used Laplace's rule, and completed basic tree diagrams. Eighth graders refreshed on the contents of the prior year's course as previously described and also calculated compound probabilities, standard deviations and variances, first degree equations with fractions, second degree equations and surfaces, and volumes of irregular prisms. Ninth graders also refreshed previous content and worked with quartiles, percentiles, and box plots, developed advanced tree diagrams, applied Laplace's rule of succession, calculated complex surfaces and volumes, and solved third- and fourth-degree equations of direct solving (using Ruffini's rule and factorizing).

### Peer tutoring implementation

During the first term, teachers for all courses used traditional teaching methods. That is, contents were taught using a one-way instructional teaching method: students could not interact

at any time, and they had to sit individually. During the second term, a peer tutoring program was implemented in combination with the teachers' lessons. Reciprocal and same-age tutoring structures were selected for this study for several reasons. On one hand, cross-age tutoring was seen as extremely difficult to implement due to bureaucratic and organizational issues. It was impossible to set several hours in which students from upper courses could tutor students from lower courses, as they had different schedules. Also, the parameters for involvement established through the legal consent were quite restrictive; for example, the tutoring program had to be implemented during school hours, and students could not leave their classes. On the other hand, previous research has shown that during fixed peer tutoring, tutees may decrease their self-concept, as they always receive help from their peers, making them feel academically less capable and not as useful as their colleagues [77, 78]. Hence, reciprocal peer tutoring was implemented so that students' roles did not negatively affect final study results.

## Classroom dynamics during peer tutoring

The classroom dynamic was as follows. First, the teacher checked students' homework, corrected that homework on the blackboard, and explained new content, which took about 15–20 minutes. Later, all students had to complete two exercises followed by either one or two problems, depending on the difficulty of the content. Students worked individually for approximately 15 minutes. The teacher helped students if they were unsure how to proceed. After that, peer tutoring was implemented for 20 minutes. Students were allowed to work in pairs, sharing their results, asking each other questions and solving the exercises and problems that they had not finished yet together. Students were told to follow the protocols and principles indicated by Topping, Buchs, Duran and Van Keer (2017) [36] including, among other issues, "pause, prompt and praise" techniques. Each teacher monitored their students' interactions. As Wingate (2019) [79] states, the teacher's role is vital in the process as he/she must ensure that students' interactions are rich in academic language and that students are effectively helping each other. Finally, during the last 5 to 10 minutes, the teacher corrected and explained the exercises and problems on the blackboard. Extra problems were given to those students who finished their work early.

## Instrument used to collect information

Students' academic self-concepts were measured using the Marsh self-concept questionnaire [80], developed by Marsh and Shavelson. This instrument is based on a Likert scale and includes reversed items. Students must grade each item from 1 (absolutely false) to 8 (absolutely true). The questionnaire contains thirteen items divided into three subscales: competence component, affective component and comparison component. Reversely coded items are included in the questionnaire such as item 3—I feel uncomfortable during mathematics class or item 7 –I'm not good at solving mathematics problems. This instrument was selected because it was specific for mathematics and its validity and reliability have been widely documented [81, 82] and because it was used in prior peer tutoring research [83, 84]. The average score for each student was used in this study. The higher the score, the higher the mathematics self-concept of the student was. Students' completed the questionnaire independently during tutoring hours and it took them between 20 to 30 minutes to complete it. A mathematics teacher was with the students while they were completing the questionnaire to solve any questions they could have about it.

## Organization and scheduling

The length of the program, number of sessions per week, and amount of time per peer tutor session were determined following indications by Leung (2015) [85] to maximize gains in

students' self-concepts. Hence, three peer tutoring sessions were held each week in all courses for the experimental group. The program lasted four weeks, and, as indicated previously, peer interactions lasted 20 minutes. It must be noted that, during the peer tutoring implementation, the control group continued with the above mentioned one-way instructional teaching method. The same teacher that did the lecture in the experimental group also did it in the control group for the same grade. Students in the experimental and control groups were provided the same problems and exercises during the development of the peer tutoring intervention.

Pairs were distributed following the suggestions provided by De Backer, Van Keer, and Valcke (2016) [86]. According to these authors, the teacher must supervise interactions between students and assist student pairs who are not able to finish the task on time. Besides, he/she also has to check the final results for each students. If both students who are paired have mistakes in their solved problems, the teacher must help them until they know how to correctly solve the problem. Students were placed in a hierarchal order from highest to lowest, according to their first-term mathematics marks. Then, in order to pair the students, the first and second students in the list constituted the first pair, the third and fourth students constituted the second pair, and so on until the list was finished. This way, the academic differences between pairs are minimized as the academic gap between tutor and tutee is the least possible. According to Matinde (2019) [87] most of the students feel very comfortable with this way of pairing as they are paired with peers whose knowledge in the subject is similar to them. Aspects such as finishing the task in a similar time or sharing similar academic goals in the subject are crucial when it comes to collaboration between students. The other main option of pairing students implied carrying out a fixed peer tutoring. In fixed peer tutoring students' are ordered by academic achievement, then the list is split in two halves. The first half are the tutors and the other half serves as tutees so that the most competent tutor is paired with the most competent tutee and so on. This way of pairing was discarded as several articles indicate that the academic gap between tutors and tutees is greater than for reciprocal peer tutoring and tutees self-concept is difficult to increase as half of the students (tutees) are almost always receiving help from their peers (tutors).

The students used the same kind of educational materials during the peer tutoring program they had used previously in the course (textbook, worksheets, and online exercises, for example). Students received training on peer tutoring development and skills days before the program was implemented. They were trained by the same teachers that taught them mathematics the whole year. Students participated actively during the training with the help of the teacher, for instance, indicating the qualities and abilities that a good tutor and a good tutee should have. Students were trained on the developing of the sessions and the nature of the interactions. The importance of sharing mathematics contents regarding the provided problems and exercises in each session and not other non-academic issues, trying to find different ways to explain a content to a tutee and valuing the different procedures used to solve a problem was highlighted. Respect and patience were defined as the basis of the interactions when working in pairs. Interactions during the tutoring sessions had to aim strictly to a shared goal: understanding and finishing all exercises and problems. First, the teacher had to check that one of the two students in each pair had solved the tasks using suitable procedures and that the final result was correct. Later, students had to share their results and procedures, checking that results were the same for both. If results did not coincide, the student who had the correct answer had to explain to the student whose answer was incorrect how to correctly solve the problem, and both had to try to detect the mistake made. Any questions regarding mathematics content were allowed during the interactions, but always from a perseverance and individual work perspective.

**Table 1. Means, standard deviations and number of students by grade, group and phase of the study.**

| | Pretest | | | | | | Posttest | | | | | |
| | Experimental | | | Control | | | Experimental | | | Control | | |
| Grade | Mean | SD | n | Mean | SD | n | Mean | SD | n | Mean | SD | n |
|---|---|---|---|---|---|---|---|---|---|---|---|---|
| 7th | 3.76 | 1.26 | 62 | 3.81 | 1.31 | 62 | 4.26 | 1.29 | 62 | 3.87 | 1.41 | 62 |
| 8th | 3.68 | 1.40 | 62 | 3.74 | 1.32 | 62 | 4.19 | 1.32 | 62 | 3.72 | 1.29 | 62 |
| 9th | 3.53 | 1.23 | 64 | 3.55 | 1.20 | 64 | 3.98 | 1.21 | 64 | 3.60 | 1.37 | 64 |
| All | 3.65 | 1.31 | 188 | 3.70 | 1.26 | 188 | 4.14 | 1.28 | 188 | 3.73 | 1.32 | 188 |

## Data analysis

SPSS software version 25 was used to analyze all student data. Means, standard deviations and gain scores were reported. Simple quantitative analysis was also carried out in order to determine the percentage of students whose self-concept scores had improved or decreased following implementation of the program. T-test (95% confidence level) was used in order to analyze the differences in gain scores between the experimental and the control group and also the differences between the posttest and the pretest scores within each group for each grade and overall [88]. Analyses of variance (ANOVAs) were used in order to analyze the differences among grades in the experimental group for the pretest scores, posttest scores and gain scores [89]. Effect sizes were calculated [90], and Hedge's *g* was provided in each case.

## Results

The descriptive results of this study are shown in Table 1, Fig 1 and Fig 2. Mean scores, standard deviations (SDs), and number of students (*n*) by grade (7th, 8th, and 9th), group (experimental or control), and phase of the study (pretest or posttest) are reported in Table 1. In order to facilitate readers' understanding of the results obtained in this study, overall scores for the experimental and control group are represented through a graph in Fig 1. Moreover, scores

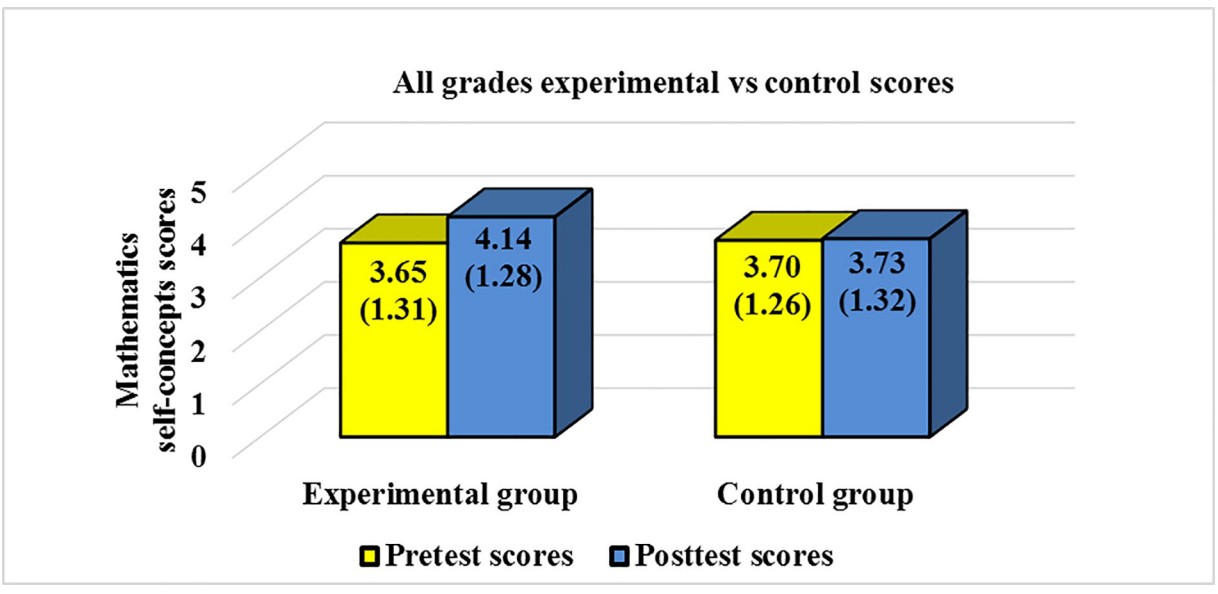

**Fig 1. Pretest and posttest overall scores and standard deviations for the experimental and control group.**

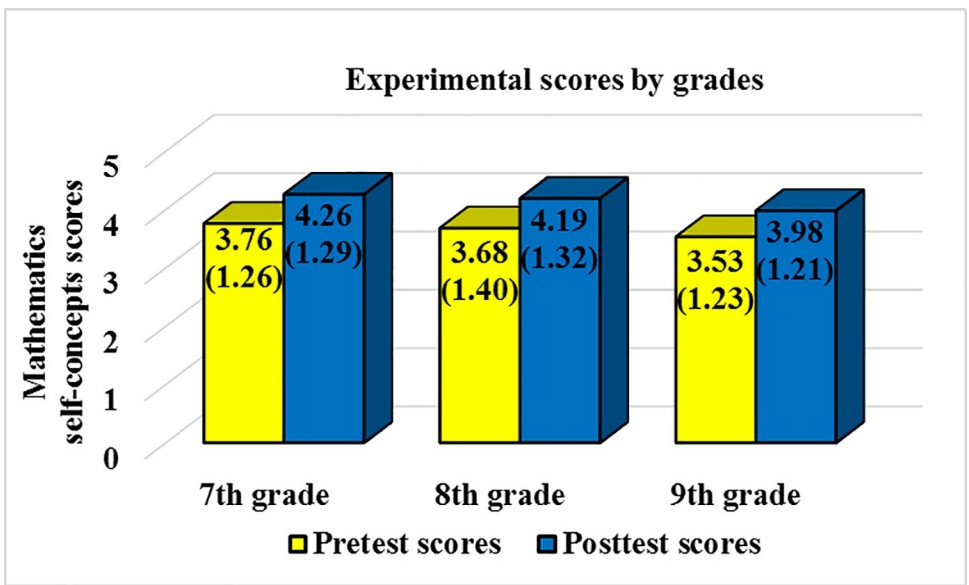

**Fig 2. Experimental group pretest and posttest scores and standard deviations by grades.**

by grade for the experimental group are represented through a graph in Fig 2. Standard deviations are included in parenthesis in both figures.

The number of students whose self-concept scores increased or decreased from the pretest to the posttest by grade and group is shown in Table 2.

The analysis of differences between the pretest and the posttest for the experimental group is shown in Table 3. Statistically significant differences between the posttest and the pretest were found individually for each grade and also overall.

The analysis of differences between the pretest and the posttest for the control group is shown in Table 4. No statistically significant differences between the posttest and the pretest were found for any of the grades nor overall.

The analysis of gain scores, that is, the difference between the posttest and the pretest scores, comparing the experimental group and the control group is shown in Table 5. Statistically significant differences were reported individually for each grade and also overall.

ANOVAS across grades were conducted for the experimental group. No statistically significant differences were reported for the pretest scores $F(2, 185) = 0.46$, $p = .63$, posttest scores $F(2, 185) = 0.02$, $p = .98$ nor gain scores $F(2, 185) = 0.55$, $p = .47$ among grades.

When analyzing the gain scores of the experimental group, an overall increment of 13.4% was found. Calculation of effect sizes showed a Hedge's $g$ value of 0.40 for 7th graders, 0.41 for 8th graders, and 0.37 for 9th graders. The global effect size for the peer tutoring program reported a Hedge's $g$ value of 0.48.

**Table 2. Number of students that increase or decrease their scores after the peer tutoring program by grades and group.**

| Grade | Experimental group | | Control group | |
|---|---|---|---|---|
| | Increase (%) | Decrease (%) | Increase (%) | Decrease (%) |
| 7th | 50 (80%) | 12 (20%) | 33 (54%) | 29 (46%) |
| 8th | 53 (86%) | 9 (14%) | 30 (48%) | 32 (52%) |
| 9th | 54 (84%) | 10 (16%) | 33 (52%) | 31 (48%) |
| All | 158 (84%) | 30 (14%) | 96 (51%) | 92 (49%) |

**Table 3. Experimental group mean differences between the posttest and the pretest and t-tests by grades and overall.**

| Grade | Mean difference | t-Value |
|---|---|---|
| 7th | 0.50 | 2.08* |
| 8th | 0.51 | 1.98* |
| 9th | 0.45 | 1.99* |
| All | 0.49 | 2.91* |

*$p < .05$

## Discussion

As stated previously, significant improvements, that is, statistically significant differences between the pretest and the posttest scores for the experimental group, were revealed for all grades as a result of peer tutoring; therefore, hypothesis 1 (students' mathematics self-concepts will improve significantly as a result of peer tutoring) was confirmed. In this sense, no significant differences were reported across grades before or after implementation of the peer tutoring program. Hence, hypothesis 2 (no statistically significant differences will be found between 7th, 8th and 9th grade students' scores before and after the implementation of the peer tutoring program) was also confirmed. Recent previous research in the field is consistent with the results found in this examination. Studies conducted by Tsuei (2012) [91], Ke (2013) [92], and Toh and Kaur (2019) [93] also showed improvements in the mathematics self-concepts of students after peer tutoring. This may be attributable to students feeling more capable and valuable in mathematics as they help each other [94, 95]. Moreover, reciprocal and same-age peer tutoring makes all students feel part of the learning process, as any student may be able to explain content throughout the program [96]. Hence, it seems like this type of tutoring has a greater effect than other types of tutoring on students' self-concepts [53]. Indeed, the effect sizes reported in the study showed that the magnitude of the effect was medium [97] and that there was a considerable improvement in the students' self-concepts.

No significant differences were reported among grades. Hence, hypothesis 2 was also confirmed. This fact, also, is consistent with prior studies, including the works of Susperreguy, Davis-Kean, Duckworth and Chen (2018) [17] and Weidinger, Steinmayr and Spinath (2019) [98]. According to several authors, differences in self-concept may be found across educational levels, that is, between primary, secondary, and postsecondary education, but are more difficult to find within the same educational level [14,15].

Although results in this experience may be seen as promising, the fact that 14% of the students in the experimental group showed lower self-concept scores must be considered. According to Drago, Rheinheimer and Detweiler (2018) [99] and Sytsma, Panahon, and Houlihan (2019) [100], although peer tutoring frequently has a positive impact on students' self-concepts, there is usually a low percentage of students (about 10–15%) that do not improve academically or psychologically. This may be because some students do not like to help other peers with academic tasks, and, although the main goal of peer tutoring is to foster

**Table 4. Control group mean differences between the posttest and the pretest and t-tests by grades and overall.**

| Grade | Mean difference | t-Value |
|---|---|---|
| 7th | 0.06 | 0.23 |
| 8th | -0.02 | 0.08 |
| 9th | 0.05 | 0.21 |
| All | 0.03 | 0.21 |

**Table 5. Mean differences between experimental and control group gain scores and t-tests by grades and overall.**

| Grade | Mean difference | t-Value |
|---|---|---|
| 7th | 0.44 | 2.01* |
| 8th | 0.53 | 2.28* |
| 9th | 0.41 | 1.99* |
| All | 0.46 | 3.30* |

*$p < .05$

collaboration, the reluctance of some students is so strong that interactions are not valuable and learning between peers does not take place. Besides, when differences between pairs are very limited, such pairs do not benefit so much from the experience. Moreover, ceiling effects must also be considered as many students already showed high scores in the pretest [101]. In this sense, as Agne and Muller (2019) [102] indicate, the supervision of the teacher plays a key role during peer tutoring. Ensuring that interactions are rich from an academic perspective and that cooperation takes place among peers are keys to ensuring that most of the students are benefiting from peer tutoring [103].

Apart from the inconvenience noted, there are certain limitations that should be considered when interpreting the results of this study. First, the sample size was quite limited, and, although some researchers may not consider it trivial, it cannot be described as large, either [104, 105]. Furthermore, although as indicated above Krejcie and Morgan (1970) [73] and Johanson and Brooks (2009) [74] state that 368 were necessary, they referred to the case in which a random sampling is viable. Authors such as Edgar, Murphy and Keating (2016) [106] state that data collected through other types different from random sampling offer no guarantee of representativity. In any case, although the sample may be representative of the population of Spanish middle school students, it is not representative of students outside the country. Future research should address the efficiency of this methodology across different countries and within different educational settings [107]. In this sense, caution is required, as many variables, such as the type of students, the time of implementation of the peer tutoring program, the frequency of sessions, the type of experimental design, and many other issues may affect the final outcome significantly [108, 109]. The fact that only one middle school was selected through the sampling process must also be considered, as a comparison among different schools could add greater value to this research [110].

## Conclusion and implications for practice

The main conclusion that can be drawn from this study is that peer tutoring may be very beneficial for middle school students' mathematics self-concepts. Considering the results of this study and previous research in the field, same-age and reciprocal peer tutoring is highly recommended for those field practitioners who want to improve the self-concepts of 7th to 9th grade students (12 to 15 years old) in mathematics. Also, from an organizational point of view, same-age and reciprocal tutoring is easier to implement, as it can take place within the same classroom of students. Given the promising results of this study and considering the previous studies in the field, low duration tutoring programs should be scheduled for four weeks or less with two to four sessions of 25 minutes or less per week for each tutoring session. The use of a control group is also highly encouraged, as its absence may result in an overestimation of effect sizes. Moreover, as a document in the literature, practitioners in the field may find improvements not only in the mathematics self-concept variable, but also in other academic and psychological variables, such as anxiety or attitude towards mathematics.

## Author Contributions

**Conceptualization:** Lidon Moliner.

**Formal analysis:** Francisco Alegre.

**Investigation:** Francisco Alegre.

**Methodology:** Francisco Alegre.

**Resources:** Lidon Moliner.

**Software:** Lidon Moliner.

**Visualization:** Francisco Alegre.

**Writing – original draft:** Francisco Alegre.

**Writing – review & editing:** Francisco Alegre.

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
