## [Decision Letter · Decision Letter 0]

17 Dec 2019

PONE-D-19-28784

Effects of peer tutoring on middle school students’ mathematics self-concepts

PLOS ONE

Dear Dr Moliner, 

I have now received detailed comments from two reviewers in regard to the manuscript. They noted the merit of the manuscript, but they also raised serious issues (e.g., the design of experiment, data analysis, etc.) and recommended major revisions. I agree with the reviewers’ assessment of the manuscript. They have offered many detailed instructions, and I will not reiterate here. I would like to give you an opportunity to revise the manuscript. If you choose to revise, please carefully consider the reviewer comments and submit a list of responses to the comments in your cover letter.

The revision must be submitted no later than 45 days from today. To enhance the reproducibility of your results, we recommend that if applicable you deposit your laboratory protocols in protocols.io, where a protocol can be assigned its own identifier (DOI) such that it can be cited independently in the future. For instructions see: http://journals.plos.org/plosone/s/submission-guidelines#loc-laboratory-protocols

We look forward to receiving your revised manuscript.

Kind regards,

Bing Hiong Ngu, Ph.D.

Academic Editor

PLOS ONE

Journal Requirements:

2. You indicated that you had ethical approval for your study. In your Methods section, please ensure you have also stated whether you obtained consent from parents or guardians of the minors included in the study (your manuscript only mentions participants' "families") or whether the research ethics committee or IRB specifically waived the need for their consent.

3. Your ethics statement must appear in the Methods section of your manuscript. If your ethics statement is written in any section besides the Methods, please move it to the Methods section and delete it from any other section. Please also ensure that your ethics statement is included in your manuscript, as the ethics section of your online submission will not be published alongside your manuscript.

Reviewers' comments:

Reviewer's Responses to Questions

**Comments to the Author**

1. Is the manuscript technically sound, and do the data support the conclusions?

Reviewer #1: Partly

Reviewer #2: Partly

2. Has the statistical analysis been performed appropriately and rigorously? 

Reviewer #1: No

Reviewer #2: Yes

3. Have the authors made all data underlying the findings in their manuscript fully available?

Reviewer #1: No

Reviewer #2: Yes

4. Is the manuscript presented in an intelligible fashion and written in standard English?

Reviewer #1: Yes

Reviewer #2: Yes

5. Review Comments to the Author

Reviewer #1: This manuscript reported an empirical study that was conducted in a Spanish middle school with the purpose of investigating the effects of peer tutoring on students’ math self-concept. Considering the school context, the effect sizes reported in the manuscript are not small and I believe may have practical significance. However, the study has one major flaw, which is not having a student achievement related variable. The only variable that the authors looked at was math self-concept. In addition, the statistical analysis reported in the results section were not clearly written and may be redundant. Detailed comments are listed below.

Major issues

1. Some important articles are missing from the literature review part. For example, Roscoe and Chi (2007) “Understanding Tutor Learning: Knowledge-Building and Knowledge-Telling in Peer Tutors’ Explanations and Questions”, Roscoe and Chi (2007) “Tutor learning: the role of explaining and responding to questions”.

2. As the authors stated in the self-concept section, self-concept can influence academic achievement. So why was this not accounted for in the hypothesis formulation?

3. What are the potential causes of research design that affect the results? If random assignment to conditions was implemented successfully, I don’t see any problems for a pretest-posttest, experimental vs. control group design.

4. Please be specific about how students were assigned to different conditions. What did the authors mean by a probabilistic base?

5. What were the subscales for the Marsh self-concept questionnaire? I suggest providing some sample items. Also, were there reversely coded items?

6. Did the teacher do the lecture or provide more problems for students to solve in the control condition?

7. Please provide more details about the peer tutoring training that students received.

8. What if both students who were paired had mistakes in their solved problems? Did they receive help from the teacher until they knew how to correctly solve the problem?

9. Which group was the experimental group? Group A or group B?

10. Line 293, what did the three tests (test 1, 2 and 3) about?

11. I cannot understand Table 3.

12. One major issue regarding the results is that the authors did not specify the independent variables and dependent variable in each statistical test they conducted.

13. Line 315. I suggest clearly stating what significant improvements were revealed.

14. I suggest restating the hypotheses in Discussion section.

15. I did not realize that the authors were also interested in the effects of different grade levels on self-concept until I finished reading the results and discussion. So I suggest including some literature in the literature review section.

Minor issues

16. It is not clear what global effect size means in the abstract.

17. First line on page 5, I think the word primary was missing from “…during the transition from school to…”

18. The sentence from line 148 – 152 is too long and difficult to understand.

Reviewer #2: Effects of Peer Tutoring On Middle School Students’ Mathematics Self-Concept

The Marsh questionnaire was used to measure participating students’ mathematics self-concepts before and after implementation of a peer-tutoring program.

The main aim of this research was to determine the effect of peer tutoring on middle school students’ mathematics self-concepts. The study tested two hypotheses.

Hypothesis 1: Students’ mathematics self-concepts will improve significantly as a result of peer tutoring.

Hypothesis 2: Students enrolled in different grades will not show significant differences regarding mathematics self-concepts before and after peer tutoring. ( This hypothesis is not clear. I had to get to line 328 to get the point of this hypothesis. )

Relevant literature was used to discuss the impact of self-concept on the learning of mathematics. However, it would make the paper more meaningful if the authors could be more expansive how positive self-concept helps with mathematics performance. Line 101 the point that students experience important self-concept changes resulting in a lower mathematics self-concept. Line 101 is circuitous, as it does not explain why there should be changes now and what these changes could affect mathematics performance.

In the section Peer tutoring and self-concept, it would be helpful if the authors could draw upon the literature cited to provide more substantial discussion how peer tutoring improve self-concept of the tutor. In line 144, the evidence is not as conclusive as one would hope. Perhaps the authors could offer reasons why the evidence are not so conclusive and how the current authors’ work could provide data to support the effect of peer-tutoring and tutors self-concept. More hypothesising here would make a better case for the hypothesis 1.

Hypothesis 2 is not clear. Perhaps the authors could provide clearer discussion and draw the reader to the meaning of this Hypothesis.

The data collection process was discussed and easy to follow More discussion could be used to explain the formation of the tutor-tutee pair. Although the authors cited the literature (De Becker et al. ) they drew upon to construct the tutor-tutee pair, what other methods did they consider to form the tutor-tutee pair and why the work of De Becker chosen, In particular how much difference in performance should there be between each member of the pair? Only when the relevant literature has been discussed could the authors use the literature to support their selection of the tutor-tutee pair.

Did the authors conduct a pilot study to help confirm such tutor-tutee pair would be effective? Without such preparation, it is difficult to give much credence to the effects of the study.

In the current paper, the authors explained the formation of the pairs. The students were arranged according to their performance, thus the first student was paired with the second, the third with the fourth, and so on. The authors, however, did not explain how different were the students in each pair. What if the first student was an A+ student (85 marks) and the second student, also an A+ student (83 marks), one would not pair these two students as the peer tutoring process would not be helpful to either one. This may explain the finding why although peer tutoring frequently has a positive impact on students,

“there is usually a low percentage of students (about 10–15%) that do not improve academically or psychologically. This may be because some students do not like to help other peers with academic tasks, and, although the main goal of peer tutoring is to foster collaboration, the reluctance of some students is so strong that interactions are not valuable and learning between peers does not take place.” Line 338 -343

It could be that when the differences between the pairs are so limited, such pairs do not benefit from the experience. Did the authors interview such students to ascertain why they did not benefit from the experience?

Studies exploring the effects of Peer-tutoring and the self-concept of tutors are important as peer-tutoring help students develop understanding of themselves as learners and also how helping others to learn strengthen and deepen their own (i.e. tutors’) knowledge of themselves as learners. In addition, self-concept of the tutors could change if they feel that their work with the tutees was beneficial. Also, peer-tutoring and its positive effects self-concept could have a last impact on how students learn. Currently the authors are looking at how self-concept is a function of peer-tutoring. But improved self-concept could help tutors learn how to learn: by helping others.

6. PLOS authors have the option to publish the peer review history of their article (what does this mean?). If published, this will include your full peer review and any attached files.

Reviewer #1: No

Reviewer #2: No

---

## [Author Response · Author response to Decision Letter 0]

2 Jan 2020

Dear Editor and Reviewers, 

We, the authors, are thankful for your consideration of this manuscript. We really appreciate all the comments and suggestions you made in your reviews as the manuscript has been improved considerably. Please find below our responses to your comments. We hope that now you find our manuscript suitable for publication. If that is not the case, please do not hesitate to tell us what you think should be done in order to improve it.

Thank you very much,

Kind regards,

The authors

Academic Editor requirements

We ensured that PLOS ONE’s style requirements were met, including those for file naming

2. You indicated that you had ethical approval for your study. In your Methods section, please ensure you have also stated whether you obtained consent from parents or guardians of the minors included in the study (your manuscript only mentions participants' "families") or whether the research ethics committee or IRB specifically waived the need for their consent.

Now this part is clearly stated in the Methodology section (sample access subsection) 

3. Your ethics statement must appear in the Methods section of your manuscript. If your ethics statement is written in any section besides the Methods, please move it to the Methods section and delete it from any other section. Please also ensure that your ethics statement is included in your manuscript, as the ethics section of your online submission will not be published alongside your manuscript.

- In the Methodology section we indicated that Ethical requirements provided by the Ethics Committee of the Spanish National Research Council (CSIC) were met during this research.

Reviewer #1 comments:

 This manuscript reported an empirical study that was conducted in a Spanish middle school with the purpose of investigating the effects of peer tutoring on students’ math self-concept. Considering the school context, the effect sizes reported in the manuscript are not small and I believe may have practical significance. However, the study has one major flaw, which is not having a student achievement related variable. The only variable that the authors looked at was math self-concept. In addition, the statistical analysis reported in the results section were not clearly written and may be redundant.

You are absolutely right about the absence of a student achievement related variable. We, the authors, thought about this possibility before starting the research. Nevertheless, we found several legal problems when accessing students’ academic achievement in mathematics. Legal consent for the self-concept variable was not very difficult to obtain, but that was not the case for students’ academic achievement. The administration was reluctant to authorize us to use students’ academic information. We take note of your comment and we hope that we can incorporate this variable in our future studies somehow.

Detailed comments are listed below.

Major issues

1. Some important articles are missing from the literature review part. For example, Roscoe and Chi (2007) “Understanding Tutor Learning: Knowledge-Building and Knowledge-Telling in Peer Tutors’ Explanations and Questions”, Roscoe and Chi (2007) “Tutor learning: the role of explaining and responding to questions”.

Both references and other recent articles were added in the literature review part.

2. As the authors stated in the self-concept section, self-concept can influence academic achievement. So why was this not accounted for in the hypothesis formulation?

As we indicated before, we did not obtain legal consent to use the academic achievement variable in our research. Although we knew that we could not perform an analysis, we wanted to note the importance of self-concept regarding mathematics achievement. If you think we must delete this part, please, just tell us and we will do it.

3. What are the potential causes of research design that affect the results? If random assignment to conditions was implemented successfully, I don’t see any problems for a pretest-posttest, experimental vs. control group design.

Our apologies for the misunderstanding. We clarified this part. What Stigmar (2016) means is that some research designs such as pretest posttest without control group or posttest only with control may overestimate the effects of a peer tutoring experience. That’s why we wanted to note that it was important to select a pretest posttest with control group design, so that reported effects were as realistic as possible.

4. Please be specific about how students were assigned to different conditions. What did the authors mean by a probabilistic base?

We clarified this part. We indicated that half the students in each course were randomly assigned to the experimental group and the other half to the control group. There were six subgroups in each grade. A draw was performed for each grade so that three subgroups were assigned to experimental conditions and the other three to control conditions. An additional final draw was performed in some subgroups to exclude some students so that the number of students in the experimental group matched the number of students in the control group in each grade. 9 students were randomly excluded due to this procedure.

5. What were the subscales for the Marsh self-concept questionnaire? I suggest providing some sample items. Also, were there reversely coded items?

The three subscales were indicated. We also indicated the existence of reversely coded items in the questionnaire and provided some sample items.

6. Did the teacher do the lecture or provide more problems for students to solve in the control condition?

We now have indicated in the subsection “organization and scheduling” that students The same teacher that did the lecture in the experimental group also did it in the control group for the same grade. Students in the experimental and control groups were provided the same problems and exercises during the development of the peer tutoring intervention.

7. Please provide more details about the peer tutoring training that students received.

Further details regarding the peer tutoring training were provided in the “organization and scheduling” section.

8. What if both students who were paired had mistakes in their solved problems? Did they receive help from the teacher until they knew how to correctly solve the problem?

Our apologies for not including this information in the manuscript. Of course, if both students who are paired had mistakes in their solved problems, the teacher had to help them until they knew how to correctly solve the problem. This is now clearly stated in the “organization and scheduling” section.

9. Which group was the experimental group? Group A or group B?

10. Line 293, what did the three tests (test 1, 2 and 3) about?

11. I cannot understand Table 3.

Our apologies for not clarifying enough what group A and group B meant and the analysis we were carrying out in table 3. We have clarified this in the manuscript right now before table 3.

First, analysis of differences for the pretest scores between experimental and control groups were conducted for each grade (tests 1 to 3). Then, analysis of differences between the posttest and the pretest scores were conducted for the experimental group by grade and overall (tests 4 to 7) and, after that, analogous analysis were carried out for the control group (tests 8 to 11). Finally, a comparison of increments, that is, the difference between the posttest and the pretest scores for experimental and control group were carried out by grade and overall (tests 12 to 15). Group A and group B were just named like that so that the reader could understand that two subgroups or groups were being compared and which of them was the minuend and the subtrahend when calculating mean differences in table 3.

12. One major issue regarding the results is that the authors did not specify the independent variables and dependent variable in each statistical test they conducted.

The mathematics self-concept scores acts as the dependent variable and the different groups are the independent variables. This has been clearly stated in the manuscript.

13. Line 315. I suggest clearly stating what significant improvements were revealed.

This has been clearly stated.

14. I suggest restating the hypotheses in Discussion section.

Hypotheses were restated in Discussion section

15. I did not realize that the authors were also interested in the effects of different grade levels on self-concept until I finished reading the results and discussion. So I suggest including some literature in the literature review section.

Some literature regarding effect sizes in the literature review section has been included 

Minor issues

16. It is not clear what global effect size means in the abstract.

“Global effect size” was changed by “overall effect size”. We just wanted to mean that this was the effect size for all grades combined and not separately.

17. First line on page 5, I think the word primary was missing from “…during the transition from school to…”

As, in all comments above, you are absolutely right. The word primary was added. Thank you very much.

18. The sentence from line 148 – 152 is too long and difficult to understand.

This sentence was divided in three separate sentences so that readers may understand it better. Thank you very much for your review.

Reviewer #2 comments:

Effects of Peer Tutoring On Middle School Students’ Mathematics Self-Concept

The Marsh questionnaire was used to measure participating students’ mathematics self-concepts before and after implementation of a peer-tutoring program.

The main aim of this research was to determine the effect of peer tutoring on middle school students’ mathematics self-concepts. The study tested two hypotheses.

Hypothesis 1: Students’ mathematics self-concepts will improve significantly as a result of peer tutoring.

Hypothesis 2: Students enrolled in different grades will not show significant differences regarding mathematics self-concepts before and after peer tutoring. (This hypothesis is not clear. I had to get to line 328 to get the point of this hypothesis. )

Our apologies for the lack of clarity regarding hypothesis 2. We have rewritten this hypothesis and we hope that now it looks clearer for the readers.

Relevant literature was used to discuss the impact of self-concept on the learning of mathematics. However, it would make the paper more meaningful if the authors could be more expansive how positive self-concept helps with mathematics performance. 

In subsection “self-concept” we explained more how positive self-concepts helps with mathematics performance.

Line 101 the point that students experience important self-concept changes resulting in a lower mathematics self-concept. Line 101 is circuitous, as it does not explain why there should be changes now and what these changes could affect mathematics performance.

The information in this line has been rewritten so that it looks clearer for readers

In the section Peer tutoring and self-concept, it would be helpful if the authors could draw upon the literature cited to provide more substantial discussion how peer tutoring improve self-concept of the tutor. 

More substantial discussion was provided from the cited literature in this subsection.

In line 144, the evidence is not as conclusive as one would hope. Perhaps the authors could offer reasons why the evidence are not so conclusive and how the current authors’ work could provide data to support the effect of peer-tutoring and tutors self-concept. More hypothesising here would make a better case for the hypothesis 1.

As in all comments above, you are absolutely right. More reasons on why the evidence are not so conclusive were given in this part and this made a better case for hypothesis 1. Thank you very much.

Hypothesis 2 is not clear. Perhaps the authors could provide clearer discussion and draw the reader to the meaning of this Hypothesis.

As stated above, hypothesis 2 was written.

The data collection process was discussed and easy to follow

Thank you very much for your kindness.

 More discussion could be used to explain the formation of the tutor-tutee pair. 

More discussion was used in this section.

Although the authors cited the literature (De Becker et al.) they drew upon to construct the tutor-tutee pair, what other methods did they consider to form the tutor-tutee pair and why the work of De Becker chosen, In particular how much difference in performance should there be between each member of the pair? Only when the relevant literature has been discussed could the authors use the literature to support their selection of the tutor-tutee pair.

This was also discussed in the section. The other main option instead of reciprocal peer tutoring was fixed peer tutoring. Fixed peer tutoring implies a different type of pairing than reciprocal peer tutoring. All this is discussed now in the manuscript.

Did the authors conduct a pilot study to help confirm such tutor-tutee pair would be effective? Without such preparation, it is difficult to give much credence to the effects of the study.

A pilot study was carried out one year before. Although only 19 students participated in the study, promising results were found. This study was published and it is referenced in the manuscript.

In the current paper, the authors explained the formation of the pairs. The students were arranged according to their performance, thus the first student was paired with the second, the third with the fourth, and so on. The authors, however, did not explain how different were the students in each pair. What if the first student was an A+ student (85 marks) and the second student, also an A+ student (83 marks), one would not pair these two students as the peer tutoring process would not be helpful to either one. This may explain the finding why although peer tutoring frequently has a positive impact on students,“there is usually a low percentage of students (about 10–15%) that do not improve academically or psychologically. This may be because some students do not like to help other peers with academic tasks, and, although the main goal of peer tutoring is to foster collaboration, the reluctance of some students is so strong that interactions are not valuable and learning between peers does not take place.” Line 338 -343. It could be that when the differences between the pairs are so limited, such pairs do not benefit from the experience. Did the authors interview such students to ascertain why they did not benefit from the experience?

You are absolutely right. We included your reflection about what happens when differences peers are so limited. Besides, ceiling effects were also indicated as some students already had high scores in the pretest. We did not interview students, but we take note of the importance of using qualitative information to contrast these findinds for our future research. Thank you very much for improving our manuscript.

---

## [Decision Letter · Decision Letter 1]

3 Feb 2020

PONE-D-19-28784R1

Effects of peer tutoring on middle school students’ mathematics self-concepts

PLOS ONE

Dear Moliner,

I have now received two reviews of your manuscript PONE-D-19-28784R1 entitled “ Effects of peer tutoring on middle school students’ mathematics self-concepts". One of the reviewers recommended a minor revision. In light of this reviewer’s recommendation, I have determined that your manuscript requires another revision before I can consider it for publication in PLOS ONE. Please revise your manuscript and submit a list of responses to the reviewer’s comments.  

We would appreciate receiving your revised manuscript no later than 35 days from today. To enhance the reproducibility of your results, we recommend that if applicable you deposit your laboratory protocols in protocols.io, where a protocol can be assigned its own identifier (DOI) such that it can be cited independently in the future. For instructions see: http://journals.plos.org/plosone/s/submission-guidelines#loc-laboratory-protocols

We look forward to receiving your revised manuscript.

Kind regards,

Bing Hiong Ngu, Ph.D.

Academic Editor

PLOS ONE

Reviewers' comments:

Reviewer's Responses to Questions

**Comments to the Author**

1. If the authors have adequately addressed your comments raised in a previous round of review and you feel that this manuscript is now acceptable for publication, you may indicate that here to bypass the “Comments to the Author” section, enter your conflict of interest statement in the “Confidential to Editor” section, and submit your "Accept" recommendation.

Reviewer #1: (No Response)

Reviewer #2: All comments have been addressed

2. Is the manuscript technically sound, and do the data support the conclusions?

Reviewer #1: Yes

Reviewer #2: (No Response)

3. Has the statistical analysis been performed appropriately and rigorously? 

Reviewer #1: No

Reviewer #2: (No Response)

4. Have the authors made all data underlying the findings in their manuscript fully available?

Reviewer #1: (No Response)

Reviewer #2: (No Response)

5. Is the manuscript presented in an intelligible fashion and written in standard English?

Reviewer #1: (No Response)

Reviewer #2: (No Response)

6. Review Comments to the Author

Reviewer #1: I enjoyed reading this manuscript. I appreciate the authors’ effort in addressing the concerns I raised in the previous review and understand the difficulty in obtaining student achievement data. I hope the minor issues listed below could help the authors further improve their manuscript.

1. Last sentence on page 4 may have typos. “…such as motivation oron…”

2. In the section Peer tutoring and self-concept, the authors first stated research related to peer tutoring in math on self-concept, then described related research in other fields. However, the authors stated research in math again at the end of this section. I suggest enhancing the structure of this section.

3. I did not see that the authors elaborated on the peer tutoring literature on how students in different grades, although the authors responded that they did. From my point of view, such elaboration can shadow Hypothesis 2.

4. The results section is not very readable. For example, what type of statistical tests were test 1 to test 3? I suggest the authors refer to APA style guide to get some ideas about how to write a results section.

5. I raised the issue in the previous review that the analysis the authors conducted were somewhat redundant, such as the ANOVA and the t tests. However, the authors did not address this in the revision.

Reviewer #2: (No Response)

7. PLOS authors have the option to publish the peer review history of their article (what does this mean?). If published, this will include your full peer review and any attached files.

Reviewer #1: No

Reviewer #2: No

---

## [Author Response · Author response to Decision Letter 1]

13 Feb 2020

Dear Editor and Reviewer, 

Thank you very much for all the work you are putting in to improve our manuscript. As you suggested, we addressed the minor revisions of one of the reviewers and we hope that our manuscript is now suitable for publication. Our responses to these revisions may be found below.

If you think we must do any additional changes or we must address any other issues, please do not hesitate to contact us,

Thank you very much,

Kind regards,

The authors

Reviewer #1 comments:

I enjoyed reading this manuscript. I appreciate the authors’ effort in addressing the concerns I raised in the previous review and understand the difficulty in obtaining student achievement data. I hope the minor issues listed below could help the authors further improve their manuscript.

Thank you very much for your thorough reviews that are helping us so much to improve our manuscript.

1. Last sentence on page 4 may have typos. “…such as motivation oron…”

Our apologies for that. We accidentally deleted a space between “or” and “on” and it became a typo. This was corrected and now it says what it was intended to say “such as motivation or on task behavior”.

2. In the section Peer tutoring and self-concept, the authors first stated research related to peer tutoring in math on self-concept, then described related research in other fields. However, the authors stated research in math again at the end of this section. I suggest enhancing the structure of this section.

You are absolutely right. We restructured this section by dividing all the information that was on it into two different sections. The first one called “peer tutoring and self-concept” now addresses experiences in subjects different of mathematics. The second one called ““peer tutoring and mathematics self-concept” specifically addresses peer tutoring and mathematics self-concept experiences.

3. I did not see that the authors elaborated on the peer tutoring literature on how students in different grades, although the authors responded that they did. From my point of view, such elaboration can shadow Hypothesis 2.

Sorry for that. We included the following information at the end of the “peer tutoring” section. Academic and psychological effects of peer tutoring may differ significantly across educational levels. For example, academic effects are usually greater in primary education than in secondary education [47, 48]. Nevertheless, effects within the same educational level are expected to be similar and, when analyzing differences across grades, significant differences are rarely reported [49, 50].

4. The results section is not very readable. For example, what type of statistical tests were test 1 to test 3? I suggest the authors refer to APA style guide to get some ideas about how to write a results section.

We indicated that tests 1 to 3 refer to analysis of differences for pretest scores between experimental and control groups for each grade. We also looked for APA style guides to report results. APA guide indicates that inferential statistical tests, tables and figures must be included and that effect sizes must be reported. As there were no figures in our manuscript, we included two figures (figure 1. Experimental vs control group overall scores and figure 2. Experimental scores by grades) so that readers of the manuscript could understand better the results section.

5. I raised the issue in the previous review that the analysis the authors conducted were somewhat redundant, such as the ANOVA and the t tests. However, the authors did not address this in the revision.

In the first ANOVA we addressed the differences between the pretest scores for the experimental group across grades and in the second ANOVA we addressed the differences in the increments for the experimental group across grades. These analysis had not been previously performed in the previous t tests, that’s why we did not delete any of these analysis. However, if you think we must delete any of these analysis, please just tell us and we will do it.

---

## [Editor Report · Decision Letter 2]

18 Feb 2020

PONE-D-19-28784R2

Effects of peer tutoring on middle school students’ mathematics self-concepts

PLOS ONE

Dear Dr Moliner,

I have now received a review of your revised manuscript PONE-D-19-28784R2 titled “Effects of peer tutoring on middle school students’ mathematics self-concepts. I concur with the reviewer that several aspects of the issues raised has not been adequately addressed. More work is needed to refine the following:

There is a need to provide a more readable presentation of the data analysis using APA guidelines (e.g., the use of APA format to report results). The information in the Tables (e.g., there are redundant words, the layout is awkward) and Figures needs rewording to improve clarity.An analysis of the difference between the experimental group and the control group for the pretest is redundant given that their mean scores only differ slightly.Greater clarity is needed to describe Test 1 and Test 3.The use of ANOVA and the t tests in the data analysis is unclear.

I want to give you an opportunity to revise your manuscript. Your manuscript could be reconsidered for publication if you can successfully attend to the issues raised. When preparing your revised manuscript, you are asked to carefully consider the comments, and submit a list of responses to the comments.  

We would appreciate receiving your revised manuscript no later than 35 days from today.  To enhance the reproducibility of your results, we recommend that if applicable you deposit your laboratory protocols in protocols.io, where a protocol can be assigned its own identifier (DOI) such that it can be cited independently in the future. For instructions see: http://journals.plos.org/plosone/s/submission-guidelines#loc-laboratory-protocols

We look forward to receiving your revised manuscript.

Kind regards,

Bing Hiong Ngu, Ph.D.

Academic Editor

PLOS ONE

---

## [Author Response · Author response to Decision Letter 2]

28 Feb 2020

Dear Editor and Reviewer, 

Thank you very much for all the work you are putting in to improve our manuscript. As you suggested, we addressed the minor revisions of one of the reviewers and we hope that our manuscript is now suitable for publication. Our responses to these revisions may be found below.

If you think we must do any additional changes or we must address any other issues, please do not hesitate to contact us,

Thank you very much,

Kind regards,

The authors

Editor comments:

There is a need to provide a more readable presentation of the data analysis using APA guidelines (e.g., the use of APA format to report results). The information in the Tables (e.g., there are redundant words, the layout is awkward) and Figures needs rewording to improve clarity.

We remade the entire results section, following APA format to report results, deleting redundant words and trying to provide a more suitable layout. Figures were also reworded in order to improve clarity.

An analysis of the difference between the experimental group and the control group for the pretest is redundant given that their mean scores only differ slightly.

We deleted this analysis

Greater clarity is needed to describe Test 1 and Test 3.

Test 1 and test 3 referred to the analysis above indicated. As you suggested, we deleted them.

The use of ANOVA and the t tests in the data analysis is unclear.

We tried to provide a clearer explanation about the use of ANOVAs and t tests in the “data analysis” section.

Reviewer #1 comments:

I enjoyed reading this manuscript. I appreciate the authors’ effort in addressing the concerns I raised in the previous review and understand the difficulty in obtaining student achievement data. I hope the minor issues listed below could help the authors further improve their manuscript.

Thank you very much for your thorough reviews that are helping us so much to improve our manuscript.

1. Last sentence on page 4 may have typos. “…such as motivation oron…”

Our apologies for that. We accidentally deleted a space between “or” and “on” and it became a typo. This was corrected and now it says what it was intended to say “such as motivation or on task behavior”.

2. In the section Peer tutoring and self-concept, the authors first stated research related to peer tutoring in math on self-concept, then described related research in other fields. However, the authors stated research in math again at the end of this section. I suggest enhancing the structure of this section.

You are absolutely right. We restructured this section by dividing all the information that was on it into two different sections. The first one called “peer tutoring and self-concept” now addresses experiences in subjects different of mathematics. The second one called ““peer tutoring and mathematics self-concept” specifically addresses peer tutoring and mathematics self-concept experiences.

3. I did not see that the authors elaborated on the peer tutoring literature on how students in different grades, although the authors responded that they did. From my point of view, such elaboration can shadow Hypothesis 2.

Sorry for that. We included the following information at the end of the “peer tutoring” section. Academic and psychological effects of peer tutoring may differ significantly across educational levels. For example, academic effects are usually greater in primary education than in secondary education [47, 48]. Nevertheless, effects within the same educational level are expected to be similar and, when analyzing differences across grades, significant differences are rarely reported [49, 50].

4. The results section is not very readable. For example, what type of statistical tests were test 1 to test 3? I suggest the authors refer to APA style guide to get some ideas about how to write a results section.

Following suggestions given by the editor, tests 1 to 3 were deleted as they were not necessary. We also looked for APA style guides to report results. APA guide indicates that inferential statistical tests, tables and figures must be included and that effect sizes must be reported. As there were no figures in our manuscript, we included two figures (figure 1. Experimental vs control group overall scores and figure 2. Experimental scores by grades) so that readers of the manuscript could understand better the results section.

5. I raised the issue in the previous review that the analysis the authors conducted were somewhat redundant, such as the ANOVA and the t tests. However, the authors did not address this in the revision.

The editor raised the same issue as you did. Following your recommendations, we remade the entire results section, following APA style, trying to make it more readable and deleting redundant analysis. We hope that this section looks much better now.

---

## [Editor Report · Decision Letter 3]

5 Mar 2020

PONE-D-19-28784R3

Effects of peer tutoring on middle school students’ mathematics self-concepts

PLOS ONE

Dear Dr Moliner,

While you have addressed other issues raised by Reviewer 1, the issue about data analysis and presentation remain a serious concern. The following need more work:

The titles of Tables and Figures need to be more explicit so as to reflect the content of Tables and Figures.There is still a lot of redundant information in the tables. For example, the pre-test and post-test in Table should appear only once rather than multiple times in the table (see Ngu & Phan, 2017, page 888).What do you mean by students’ t-test? Researchers normally use a t-test to examine the mean difference between two groups. I never heard of a student’s t-test.The analysis of ANOVA is very confusing. I suggest you re-analyse the data. You can either perform a repeated measures ANOVA or t-test. You may perform a 2 (group: experimental vs. control) x 2 (test: pre-test vs. post-test) ANOVA for each grade separately and then for the all grades combined. The group is a between-group factor, and the test is a repeated measures within-subject factor. Alternatively, perform a t-test between the experimental group and the control group on gain score (i.e., post-test – pre-test) for each grade separately and then for all grades.

I want to give you yet another opportunity to revise your manuscript. If you can successfully address the issues related to data presentation and analysis, your manuscript could be reconsidered for publication in PLOS ONE.

We would appreciate receiving your revised manuscript not later than 35 days from today. To enhance the reproducibility of your results, we recommend that if applicable you deposit your laboratory protocols in protocols.io, where a protocol can be assigned its own identifier (DOI) such that it can be cited independently in the future. For instructions see: http://journals.plos.org/plosone/s/submission-guidelines#loc-laboratory-protocols

We look forward to receiving your revised manuscript.

Kind regards,

Bing Hiong Ngu, Ph.D.

Academic Editor

PLOS ONE

---

## [Author Response · Author response to Decision Letter 3]

18 Mar 2020

Reviewer #1 comments:

I enjoyed reading this manuscript. I appreciate the authors’ effort in addressing the concerns I raised in the previous review and understand the difficulty in obtaining student achievement data. I hope the minor issues listed below could help the authors further improve their manuscript.

Thank you very much for your thorough reviews that are helping us so much to improve our manuscript.

1. Last sentence on page 4 may have typos. “…such as motivation oron…”

Our apologies for that. We accidentally deleted a space between “or” and “on” and it became a typo. This was corrected and now it says what it was intended to say “such as motivation or on task behavior”.

2. In the section Peer tutoring and self-concept, the authors first stated research related to peer tutoring in math on self-concept, then described related research in other fields. However, the authors stated research in math again at the end of this section. I suggest enhancing the structure of this section.

You are absolutely right. We restructured this section by dividing all the information that was on it into two different sections. The first one called “peer tutoring and self-concept” now addresses experiences in subjects different of mathematics. The second one called ““peer tutoring and mathematics self-concept” specifically addresses peer tutoring and mathematics self-concept experiences.

3. I did not see that the authors elaborated on the peer tutoring literature on how students in different grades, although the authors responded that they did. From my point of view, such elaboration can shadow Hypothesis 2.

Sorry for that. We included the following information at the end of the “peer tutoring” section. Academic and psychological effects of peer tutoring may differ significantly across educational levels. For example, academic effects are usually greater in primary education than in secondary education [47, 48]. Nevertheless, effects within the same educational level are expected to be similar and, when analyzing differences across grades, significant differences are rarely reported [49, 50].

4. The results section is not very readable. For example, what type of statistical tests were test 1 to test 3? I suggest the authors refer to APA style guide to get some ideas about how to write a results section.

Following suggestions given by the editor, tests 1 to 3 were deleted as they were not necessary. We also looked for APA style guides to report results. APA guide indicates that inferential statistical tests, tables and figures must be included and that effect sizes must be reported. As there were no figures in our manuscript, we included two figures (figure 1. Experimental vs control group overall scores and figure 2. Experimental scores by grades) so that readers of the manuscript could understand better the results section.

5. I raised the issue in the previous review that the analysis the authors conducted were somewhat redundant, such as the ANOVA and the t tests. However, the authors did not address this in the revision.

The editor raised the same issue as you did. Following your recommendations, we remade the entire results section, following APA style, trying to make it more readable and deleting redundant analysis. We hope that this section looks much better now.

---

## [Editor Report · Decision Letter 4]

24 Mar 2020

Effects of peer tutoring on middle school students’ mathematics self-concepts

PONE-D-19-28784R4

Dear Dr. Moliner,

I am pleased to inform you that I am accepting your manuscript PONE-D-19-28784R4 "Effects of peer tutoring on middle school students’ mathematics self-concept" for publication in PLOS ONE.

With kind regards,

Bing Hiong Ngu, Ph.D.

Academic Editor

PLOS ONE
---

## [Editor Report · Acceptance letter]

26 Mar 2020

PONE-D-19-28784R4 

Effects of peer tutoring on middle school students’ mathematics self-concepts 

Dear Dr. Moliner:

I am pleased to inform you that your manuscript has been deemed suitable for publication in PLOS ONE. Congratulations! Your manuscript is now with our production department. 

With kind regards,

on behalf of

Dr. Bing Hiong Ngu 

Academic Editor

PLOS ONE